# Efficacy of an Educational Intervention for Sodium Restriction in Patients with Hypertension: A Randomized Controlled Trial

**DOI:** 10.3390/nu15092159

**Published:** 2023-04-30

**Authors:** Marcela P. Rodrigues, Carolina B. Ferreira, Kauane Aline M. Dos Santos, Paula N. Merello, Sinara L. Rossato, Sandra C. Fuchs, Leila B. Moreira

**Affiliations:** 1Postgraduate Studies Program in Cardiology, Universidade Federal do Rio Grande do Sul, Porto Alegre 90035-002, Brazil; 2Nursing School, Universidade Federal do Rio Grande do Sul, Porto Alegre 90620-110, Brazil; 3Medicine School, Universidade Federal do Rio Grande do Sul, Porto Alegre 90035-002, Brazil; 4Graduation Course in Collective Health, Institute of Geography, Universidade Federal de Uberlandia (UFU), Uberlandia 38400-902, Brazil; 5Hospital de Clínicas de Porto Alegre, Porto Alegre 90035-903, Brazil

**Keywords:** sodium restriction, educational intervention, dietary sodium restriction, low-sodium diet, adherence, hypertension, Dietary Sodium Restriction Questionnaire

## Abstract

There is sound evidence showing the efficacy of non-pharmacological interventions in lowering blood pressure (BP); however, adherence is usually poor. Interventions to induce behavioral changes aim to improve the ability to read labels, choose foods, and eat low-sodium meals, reinforcing adherence to sodium restriction. In this randomized parallel-controlled trial, we assessed the effectiveness of an educational intervention using the Dietary Sodium Restriction Questionnaire (DSRQ) scores. A follow-up period of 6 months was conducted. Participants were randomized into (1) an educational intervention provided by a registered dietitian on individual visits and dietary planning; (2) a control group with the usual care and dietary recommendations. Patients underwent 24-h ambulatory BP monitoring, 12-h fasting blood tests, spot urine collection, and assessment using DSRQ. We randomized 120 participants (67.5% women and 68.3% Caucasians), and 25 participants were lost to follow-up. The 24-h sodium urinary excretion changed in the control (Δ −1610 mg/day; 95% confidence interval [CI] −1800 to −1410) and intervention groups (Δ −1670 mg/day; 95% CI −1800 to −1450) over time. There was no significant difference in the 24-h estimated sodium between groups. In hypertensive patients, DSRQ-based educational intervention is effective for improving the ability to detect and overcome obstacles to a low-sodium restriction diet but is as effective as dietary recommendations for lowering sodium.

## 1. Introduction

High blood pressure (BP) is the primary preventable cause of mortality and disability worldwide, resulting in over 10.4 million deaths and 218 million disability-adjusted life-years [1]. Guidelines for managing hypertension recommend non-pharmacological interventions as the first-line strategy, with a focus on lifestyle changes [2,3,4,5]. Specifically, these interventions emphasize increasing physical activity, dietary modifications, and weight loss [6]. The DASH diet (Dietary Approaches to Stop Hypertension) [7] emphasizes the consumption of fruits, vegetables, whole grains, and low-fat dairy products while discouraging fast foods, fried foods, and red meats, and has been shown to lower BP levels. However, a subsequent study on implementing the DASH diet with a restriction on high-sodium foods resulted in an even greater reduction in BP [8]. As a result, there has been an increasing emphasis on reducing sodium intake in the recommendations for both normotensive and hypertensive individuals [9,10].

Despite recommendations to cook with low sodium and prefer fresh foods, the average Brazilian population consumes 4700 milligrams of sodium per day [11], more than double the recommended daily amount [12]. While it has been well-established in both experimental [7,8] and observational [13,14] settings that sodium restriction can effectively lower BP, food consumed away from home, which typically contains high amounts of sodium [13], can be a challenging task for many individuals. There are several potential explanations for the challenges associated with reducing sodium intake including a lack of information about the risks of high BP related to sodium intake, the impact of sodium reduction on taste, and the difficulty of accurately estimating the amount of sodium in food and culinary preparations [15,16]. Interventions based on the Theory of Planned Behavior (TPB) [17] have proven useful in guiding initiatives aimed at promoting healthy eating habits [17,18]. This theory has been widely applied to study behaviors in various fields, including advertising, public relations, and health care, to study human behaviors.

The TPB emphasizes that a person’s behavioral intention is the most crucial factor, with attitude, subjective norm, and perceived behavioral control being the determinants of behavioral intention. The validation of the Dietary Sodium Restriction Questionnaire (DSRQ) as a valuable tool [19,20] has enabled its use in identifying the factors influencing compliance with recommendations for a low-sodium diet. Therefore, in this study, we utilized a validated, adapted, and internally consistent version of the DSRQ [21], to evaluate attitudes and barriers associated with sodium restriction among patients with hypertension in southern Brazil. The DSRQ [21] comprises three subscales that correspond to the determinants of the TPB: attitude and subjective norm, perceived behavioral control, and dependent behavior. The attitude and subjective norm subscale assess patients’ beliefs about the outcomes of executing the behavior and their motivation to comply with the beliefs of significant others. The perceived behavioral control subscale evaluates patients’ ability to identify barriers and facilitators related to the behavior. The dependent behavior subscale is related to decision-making situations for food choices outside of their home. A better understanding of the factors that regulate adherence to dietary sodium restriction should be a primary goal of interventions that can be adequately planned and implemented [20]. Our study hypothesized that patients with hypertension who participated in the intervention group would show improved adherence to a low-sodium diet, resulting in decreased BP values assessed by 24-h ambulatory monitoring (ABPM) and urinary sodium excretion values after three and six months compared to the control group. Therefore, the aim of our study was to determine the efficacy of an educational intervention based on DSRQ scores in reducing sodium consumption and lowering BP.

## 2. Materials and Methods

This parallel randomized controlled trial included participants from the general population and the Hypertension outpatient clinic at Hospital de Clínicas de Porto Alegre (HCPA). Prior to enrollment, participants provided informed consent, and the study protocol (protocol number 150496) was approved by the hospital’s institutional review board. Additionally, the study protocol has been registered at ClinicalTrials.gov (NCT02848690) and published [22].

### 2.1. Participants and Randomization

Participants were recruited from the outpatient clinic or through media advertisements. Potentially eligible participants were screened and those who met the criteria were invited to a clinical visit. Eligible participants were men and women aged 40 to 80 years who had a diagnosis of hypertension and were undergoing BP-lowering drug treatment at the clinic and had not been visiting a nutritionist in the previous six months. Participants were invited to participate and, after signing a consent form, were randomly allocated to either the educational intervention or the control group. Exclusion criteria included gastrointestinal tract disease, inflammatory disease, chemotherapy treatment, physician diagnosis of diabetes mellitus, those with cognitive impairment detected during the interview, and those unable to participate without third-party involvement, pregnant or lactating women. The follow-up was carried out for up to six months with monthly monitoring.

### 2.2. Randomization, Allocation Concealment, and Blinding

A randomization list was generated using software (randomization.com) at a 1-to-1 ratio, with participants allocated in blocks of six, and the randomization sequence was created by an independent researcher outside the clinic. The sequence was stored in opaque sealed envelopes which were kept outside the clinical center for blinding purposes.

Participants and the research team were not blinded to the intervention and control groups. However, the assessment of outcomes was conducted by an independent blinded researcher.

Between November 2015 and October 2017, we screened 460 participants. Out of these, 160 did not meet the eligibility criteria, and 180 declined to participate. A total of 120 participants were randomized to the intervention and control groups. By April 2018, 15 and 10 participants were lost to follow-up in the intervention and control groups, respectively. Figure 1 illustrates the screening and enrollment process.

### 2.3. Educational Intervention Group

A registered dietitian oversaw the educational intervention and provided participants in the intervention group with detailed guidance and recommendations on adhering to a low-sodium diet. During their initial consultation, the participants received a dietary plan that emphasized the consumption of fruits, vegetables, low-fat dairy, and low-fat and minimally processed food. Additionally, their daily energy intake was reduced by 500 to 1000 kcal based on their baseline weight. To ensure continued adherence to the low-sodium diet, the participants attended monthly 60-min face-to-face sessions with the registered dietitians who provided encouragement and motivation.

The DSRQ [21] was applied at the baseline to guide intervention activities and strategies, after three months, and at the end of the follow-up to evaluate participant performance concerning attitudes and barriers associated with sodium restriction. The intervention approach involved empowering individuals to achieve their goals and make behavioral changes to adhere to a low-sodium diet. Progress was tracked to identify skill acquisition and overcome barriers to adherence. Participants developed personalized activities and strategies based on their DSRQ subscale results to increase adherence. These included learning sessions to understand the significance of a low-sodium diet and provide guidance on food choices, such as reading labels, selecting and cooking low-sodium items, choosing suitable restaurant options, and adjusting food and flavor preferences. Dietary intake was assessed using three-day food records to monitor diet adherence.

The activities were developed according to the instrumental subscales: (i) attitude and subjective norm, participants received explanations to understand the low-sodium dietary importance to control hypertension and the influence of family and others in choices and food preparation; (ii) perceived behavioral control, they identified barriers to low sodium adherence, such as lack of knowledge, interference with socialization, and lack of food selections; they learned to increase information about food choices, cooking or preparing food without sodium, low-sodium diet shopping, evaluating recipes and making suggestions for changes to low-sodium food, and to read labels; (iii) dependent behavior, they received learning sessions on the amount of sodium in food, including sodium quantity demonstrations, low-sodium food selection, changes in food choices in restaurants, and being aware of the need to change both taste and food preferences.

The educational intervention was individualized and based on dietary information provided by the patient. They answered the 24-h dietary recall for monitoring low-sodium diet adherence. The 24-h dietary recall was analyzed by computer software, the DietSys [23], to calculate the sodium content of the patient’s dietary intake. Both groups were followed up in face-to-face appointments monthly, for six months and requested to not change their physical activity practice.

### 2.4. Control Group

Participants allocated to the control group had monthly appointments with registered dietitians, in line with standard care practices. During the first visit, participants received an explanatory leaflet on hypertension and provided general recommendations. These recommendations included reducing sodium intake, avoiding high-sodium foods and alcoholic beverages, and losing weight if a body mass index was greater than 25 kg/m^2^. The DSRQ [22] was administered at baseline, three months after the randomization, and at the end of the follow-up period. Anthropometric and office BP measurements were performed monthly during 30-min face-to-face appointments throughout the follow-up period.

Anthropometric and office BP measurements were performed monthly. The 24-h ambulatory BP monitoring was performed at baseline and at the end of the study.

Instruments used for data collection: The DSRQ has been validated for patients with hypertension [21] and comprises 26 items. The initial section of the instrument comprises 11 descriptive, multiple-choice items, while the second section features three subscales. Each question in the second section is scored on a Likert scale and the three subscales include the attitude and subjective norm subscale, perceived behavioral control subscale, and dependent behavior subscale.

The subscale scores for attitude and subjective norm vary from 9 to 40, where higher scores indicate a more positive attitude toward a low-sodium diet and greater motivation to follow it due to the approval of significant others. On the other hand, the scores for perceived behavioral control range from 3 to 15, where a higher score indicates a lower perceived ability to stick to a low-sodium diet. Finally, the scores for dependent behavior subscale range from 4 to 20, where a higher score signifies greater challenges in adhering to the low-sodium diet. Trial management:

The research team comprised trained graduate students from the Nursing and Medicine Schools, as well as two registered dietitians who were postgraduate students. Their responsibilities included recruiting participants, inviting them to participate, administering informed consent, screening for eligibility, and collecting data. Additionally, the registered dietitians applied the DSRQ.

The collection of data took place at the Clinical Research Center located at HCPA. To measure BP, ABPM was conducted at the beginning and end of the six-month intervention period using Spacelabs 90207 devices (Redmond, WA, USA). BP was recorded every 15 min from 6 am to 10 pm and every 20 min during nighttime. The validity of ABPM required at least 16 daytime and 8 nighttime readings [2]. Clinical assessments, including total cholesterol, HDL cholesterol, LDL cholesterol, triglycerides, glucose, and potassium, were taken from a 12-h fasting blood sample at baseline and follow-up. Additionally, spot urine samples were collected at baseline, two months, and the end of the study to measure sodium excretion and estimate dietary intake. The DSRQ was administered to both groups at baseline, after three months, and after six months of implementation. Participants were instructed to maintain their physical activity levels throughout the study. At each visit, participants underwent two standardized BP measurements [3] using validated oscillometry (Omni 612 Model), and the average was recorded. We also obtained standardized anthropometric measurements in duplicate, including body weight (kg), height (cm; Lider^®^ model P200C), and waist circumference (cm). Three sets of 24-h dietary recalls were obtained for low-sodium diet adherence monitoring, detailing the food and portion consumed and including information on time, place, and quantity. All research assistants had previous experience with the instrument and were certified to perform the measurements.

### 2.5. Outcomes

The outcomes were calculated based on the difference between follow-up and baseline estimated 24-h sodium consumption, and systolic and diastolic BP (intervention delta minus control delta) assessed by ABPM. The 24-h sodium excretion was estimated from a spot of urine using a simple formula with high sensitivity to detect patients with sodium intake greater than 3600 mg/day [24]. The formula to estimate the 24-h sodium excretion was developed from multivariate regression equation coefficients. Women: Estimated 24-h urinary sodium excretion (g/day) = 0.15 + (weight in kg × 0.03) + (sodium in the urine specimen in g/L × 0.63).

Men: Estimated 24-h urinary sodium excretion (g/day) = 0.96 + (weight in kg × 0.03) + (sodium in the urine specimen in g/L × 0.63).

### 2.6. Statistical Analysis

To determine the appropriate sample size, the average sodium intake of 3900 ± 1602 mg/day (168.5 mmol) [23] was taken into account. The study aimed to detect a difference of 2300 mg/day (100 mmol) between the two groups. A sample with 48 participants per group had 80% statistical power to detect an estimated urinary sodium difference of at least 40 mmol (±69.2) between groups with an alpha error of 0.05. To account for potential losses in the follow-up, we enrolled 60 participants per group.

The characteristics of the sample were presented as mean with standard deviation or absolute numbers and percentages. Group comparisons were conducted using Pearson’s chi-square test for categorical variables and Student’s *t*-test for continuous variables. The efficacy of the educational intervention to reduce sodium consumption was analyzed using the change in the 24-h sodium estimated means. Groups were compared using a generalized estimating equation (GEE) adjusted for baseline systolic and diastolic BP and the use of BP-lowering drugs. The changes in urinary sodium, systolic BP, diastolic BP, and weight were analyzed using a general linear model adjusted for urinary sodium, systolic and diastolic BP, and baseline weight. The changes in the DSRQ subscales were compared between groups using GEE.

The analysis was conducted following the intention-to-treat principle and utilized PASW Statistics 18^®^ (International Business Machines Corp., New York, NY, USA). Because the number of participants included in each analysis is not equal due to missing data, the n was indicated according. Differences with *p*-value < 0.05 were considered statistically significant, and 0.05 < *p*-value < 0.10 as a trend toward association.

## 3. Results

Out of the 120 participants initially enrolled, 45 individuals completed the follow-up in the intervention group, and 50 completed it in the control group. The majority of the participants were women (67.5%) and Caucasian (68.3%), with a mean age of 61.8 ± 10.0 years. Additionally, both groups had a similar duration of hypertension diagnosis, with approximately 18 ± 13 years.

The intervention and control groups had similar characteristics; approximately two-thirds had prior dietary counseling beyond six months, most reported receiving advice to reduce sodium intake, and 30% were currently following a low-sodium diet. At baseline, most participants did not go to restaurants, preferring to cook and eat at home, bought meats, cereals, fruits, and vegetables that were not ready to eat, and did not consume ultra-processed foods (Table 1).

During the follow-up period, attitude and subjective norm subscale scores increased in the intervention group compared to the control, showing that participants agreed with the benefits of a low-sodium diet and motivation enhancement following the approval of significant others (*p* = 0.04). Throughout the follow-up period, the intervention group exhibited a significant increase in attitude and subjective norm subscale scores compared to the control group (*p* = 0.04), indicating that participants in the intervention group were more likely to perceive the benefits of a low-sodium diet and feel motivated by the approval of significant others. The lower score on the perceived behavioral control subscale (*p* = 0.02) suggests that participants in the intervention group had less difficulty identifying barriers to adherence to the sodium-restricted diet and more ability to adhere to the low-sodium diet (Table 2).

Table 3 displays the delta differences in 24-h estimated sodium intake, BP, and weight between the intervention and control groups. The results indicate that there were no statistically significant differences in estimated 24-h sodium intake or BP measurements obtained through ABPM between the two groups. There was no significant difference between groups for estimated 24-h urinary sodium (*p* for interaction = 0.80), even adjusting for baseline systolic and diastolic BP and BP-lowering drugs (Figure 2). The reduction in per-group sodium intake, as estimated by urinary sodium excretion, was found to be significant in both groups (*p* < 0.001). Weight reduction was significant in the intervention group compared with the control group (*p* = 0.02).

## 4. Discussion

The aim of this randomized controlled trial was to compare the effectiveness of an educational intervention that used DSRQ with usual care provided by registered dietitians. After six months, the estimated sodium urinary excretion was similarly reduced in both groups. However, no statistically significant reduction was observed in 24-h ABPM measurements after six months.

For the initial time, the validated edition of the DSRQ was employed in a randomized clinical trial, specifically for hypertensive patients located in southern Brazil. At the follow-up, the DSRQ score for the attitude and subjective norm scale showed that participants in the intervention group identified more benefits of following a low-sodium diet compared to the control group. They also reported fewer barriers related to reducing sodium intake, which improved their perceived behavioral control subscale. Furthermore, the participants were incentivized to modify their conduct and acquire abilities for creating meals with reduced sodium content. In addition, the individuals expressed that they had adjusted to the flavor of low-sodium foods. However, on the subscale that measures dependent behavior, the choice of a bistro or restaurant did not affect the participants’ commitment to maintaining a low-sodium diet. On the other hand, subjects stated that they became accustomed to the taste of low-sodium foods and changed their meal preparation habits, reducing sodium and abstaining from sodium-based condiments. Participants also expressed a preference for homemade meals and buying unprocessed foods such as meats, cereals, fruits and vegetables, avoiding ultra-processed foods.

After a 6-week intervention, the results of a study [25] conducted on patients with heart failure showed an improvement in attitude subscale scores, which suggests a positive shift in their attitudes toward adhering to a low-sodium diet. Our findings agree with other studies regarding participants perceiving fewer barriers to sodium restriction, improving the ability to read labels, choose foods and eat meals, and reinforcing the perceived ability to deal with possible obstacles [26,27]. Participants presented positive attitudes toward following a low-sodium diet. The knowledge about the low-sodium diet’s importance to control hypertension may influence dietary intake after explanation.

Our results agree with a study performed on hypertensive adults and the elderly, who had an intake above the recommended, primarily based on sodium added during cooking and at the table [28]. Like our findings, other studies showed that most participants reported receiving advice about sodium restriction, following a prescribed diet, and considering a convenient diary [25,29,30]. Despite setting a sodium intake goal for patients with hypertension, the 24-h sodium estimated at the beginning of the study was found to be higher, which indicated a lack of dietary adherence. The previous study conducted on heart failure patients [25], which also employed the use of DSRQ, presented no variation in sodium intake or 24-h urinary sodium levels. This suggests that interventions that rely on the application of DSRQ may not bring about any significant changes in sodium excretion beyond what is achieved through typical care methods.

It is worth mentioning that a few participants in our study had also taken part in a prior study and had received extensive guidance on diet, sodium restriction, and lifestyle changes from different healthcare teams for a considerable length of time. As a result, the residual impact of their previous experiences may have played a role in the reduction in sodium intake observed in both groups.

Shamsi et al. reported that a lifestyle intervention in the form of a continuous care model decreased the mean dietary sodium intake and systolic and diastolic BP in hypertensive patients compared with usual care [31]. Unlike our study, their control group received no lifestyle intervention or counseling. In our study, the reduction in sodium intake did not translate to BP reduction. This population frequently uses diuretics which can influence sodium excretion [32]; however, its use was similar between groups.

Our findings corroborate previous interventions based on the Theory of Planned Behavior in Brazilian hypertensive patients to promote low-sodium intake [33]. The interventions led to a marked decrease in the total amount of sodium added to meal preparation but there was no significant decrease in the 24-h urinary excretion of sodium. In addition, behaviors linked to sodium intake are influenced by a variety of factors, including motivation (intention) and pleasure-related aspects [34]. In other words, adhering to a low-sodium diet is a multifaceted process that involves various elements, including obstacles, skills, knowledge, and the motivation to reduce sodium consumption. It is necessary to recognize the factors that impact dietary adherence, such as treatment, difficulty in modifying eating behaviors, lack of motivation, insufficient knowledge, inadequate social support, and a perceived lack of benefits [18,35]. Intervention strategies to reduce sodium should focus on attitudes and cultural norms to promote dietary change.

This study presented some limitations that deserve consideration. The collection of urine spots can be utilized to estimate sodium intake; however, its accuracy remains a topic of controversy [36,37,38,39,40]. The sodium excretion, assessed by urine spot, may not provide a complete representation of all variations in sodium intake over a period and its dependability may be influenced negatively by the use of diuretics [36], which are frequently administered to this group of people.

Nevertheless, predictive equations for estimating an individual’s mean 24-h sodium excretion may have some value in monitoring sodium intake progress, despite the limitations [37,39]. Additionally, it is worth noting that the formula [24] used to estimate sodium intake was initially developed for the Brazilian population, where there is a high prevalence of sodium intake (4700 mg per day) [11]. Moreover, the formula was designed specifically for people with renal disease and did not utilize urinary creatinine and potassium data [24]. Finally, individual loss to follow-up or even those who decided not to remain in the study occurred similarly in both groups. The overall sociodemographic characteristics in the intervention and control groups were similar to the participants who completed the follow-up.

## 5. Conclusions

In conclusion, implementing an educational intervention for sodium restriction based on the DSRQ by a multidisciplinary team, including a dietitian, improves attitude and subjective norms in patients with hypertension. This intervention also reinforces their knowledge, skills, and perception of the benefits of low-sodium diets. However, it is not enough to overcome the effect of regular advice.

## Figures and Tables

**Figure 1 nutrients-15-02159-f001:**
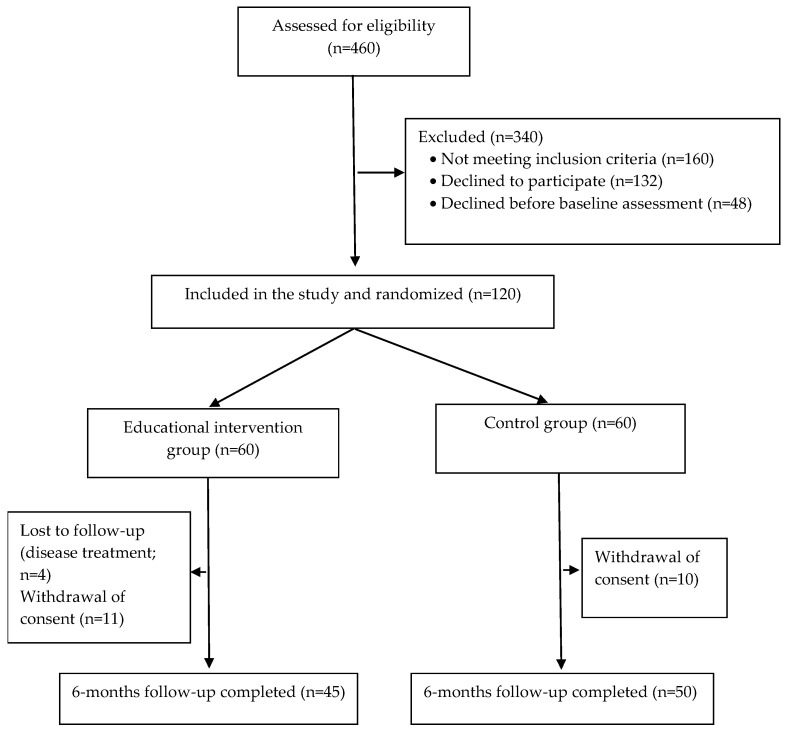
Study flowchart.

**Figure 2 nutrients-15-02159-f002:**
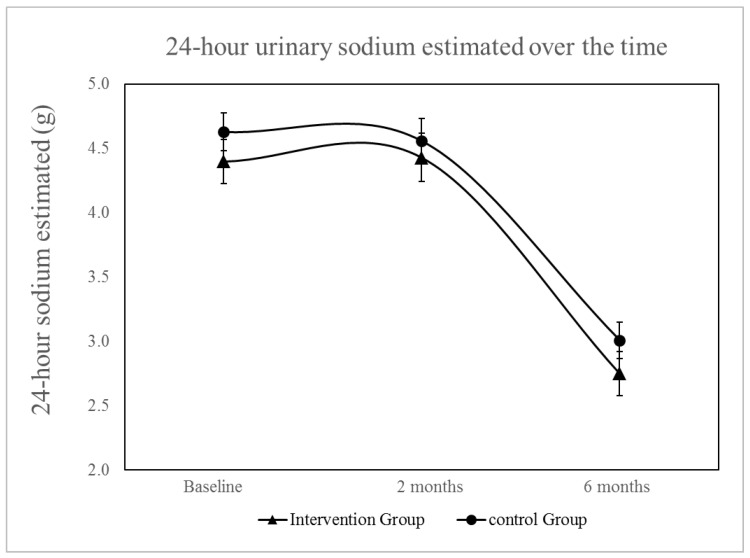
Changes in 24-h urinary sodium excretion over time, with adjustments made for baseline SBP and DBP, as well as the use of diuretics, adrenergic blockers, beta-blockers, ACE inhibitors, vasodilators, calcium channel blockers, AT1 receptor antagonists of angiotensin II. The reduction in urinary sodium excretion over time was statistically significant for within-group comparisons (*p* < 0.001) but there was no significant interaction between groups and time (*p* = 0.761).

**Table 1 nutrients-15-02159-t001:** Baseline participants’ characteristics [n (%) or mean ± SD].

	Intervention Group (n = 60)	Control Group (n = 60)	*p*-Value
Women	41 (68.3)	40 (66.7)	0.8
Caucasian	39 (65.0)	43 (71.7)	0.4
Age (years)	61.9 ± 10.2	61.7 ± 9.9	0.9
Education (years)	8.3 ± 4.2	8.0 ± 4.4	0.8
Prior dietary counseling	41 68.3%	40 66.7%	0.8
Following low-sodium diet	19 (31.7)	18 (30.0)	0.84
Do not go to restaurants	34 (56.7)	33 (55.0)	0.85
Estimated 24-h sodium intake (g/day)	4.4 ± 1.2	4.6 ± 1.1	0.5
24-h systolic blood pressure (mmHg)	123.3 ± 18.8	121.4 ± 18.0	0.6
24-h diastolic blood pressure (mmHg)	72.2 ± 11.7	73.1 ± 13.7	0.7
Waist circumference (cm)	98.5 ± 10.6	100.0 ± 20.2	0.6
Body mass index (kg/m^2^)	30.5 ± 5.4	30.4 ± 4.9	1.0
Total cholesterol (mg/dL)	190.4 ± 45.8	188.7 ± 37.9	0.8
LDL cholesterol (mg/dL)	105.1 ± 31.6	105.9 ± 35.7	0.9
HDL cholesterol (mg/dL)	53.0 ± 15.9	50.4 ± 14.0	0.4
Triglycerides (mg/dL)	171.2 ± 243.5	158.4 ± 98.2	0.7
Fasting glucose (mg/dL)	96.4 ± 13.5	96.1 ± 13.0	0.9
Creatinine (mg/dL)	0.8 ± 0.3	0.9 ± 0.3	0.7
Potassium (mEq/L)	4.5 ± 0.4	4.5 ± 0.5	0.9
Use of BP-lowering medication			
Diuretic (mg/day)	42.5 ± 61.3	55.8 ± 127.8	0.5
Adrenergic blocker (mg/day)	21.7 ± 134.0	5.2 ± 39.4	0.4
Beta-blocker (mg/day)	95.7 ± 127.3	106.9 ± 107.9	0.6
ACE inhibitor (mg/day)	38.1 ± 59.3	27.0 ± 45.6	0.3
Vasodilator (mg/day)	33.1 ± 81.1	21.1 ± 56.1	0.4
Calcium channel blocker (mg/day)	4.2 ± 7.0	9.9 ± 28.8	0.1
AT1R antagonist (mg/day)	8.6 ± 24.5	25.0 ± 44.0	0.01

ACE, angiotensin-converting enzyme; AT1R, Angiotensin II Receptor Type-1. Laboratory reference values: Total cholesterol <190 mg/dL; LDL-C <130 mg/dL; HDL-C >40 mg/dL; Triglycerides <150 mg/dL; Fasting glucose <100 mg/dL; Creatinine 0.50–1.20 women and 0.60–1.30 mg/dL men; Potassium 3.5–5.1 mEq/L.

**Table 2 nutrients-15-02159-t002:** Patient barriers to low-sodium dietary adherence, attitude, and benefit scores [mean ± SD].

	Intervention Group n = 45		Control Group n = 50		*p*-Value for Interaction
	Baseline	6-Months	Baseline	6-Months	
Subscale: attitude and subjective norm *	35.5 ± 3.7	38.3 ± 2.4	35.9 ± 5.0	37.1 ± 3.1	0.04
Subscale: perceived behavioral control **	5.3 ± 3.1	3.5 ± 1.2	4.7 ± 2.6	3.9 ± 1.6	0.02
Subscale: dependent behavior **	9.0 ± 5.2	7.2 ± 4.4	8.9 ± 4.7	7.5 ± 3.3	0.4

* Higher score indicates a more positive attitude toward a low-sodium diet. ** Higher score indicates greater difficulty in adhering to the low-sodium diet.

**Table 3 nutrients-15-02159-t003:** Delta differences for 24-h sodium estimated, blood pressure, and weight, index between groups.

Variables n = Intervention/n = Control *	Intervention Group Δ (95%CI)	Control Group Δ (95%CI)	Differences between Groups Δ (95%CI) **	*p*-Value for Interaction Interaction ***
24-h sodium estimated (g/day) 44/46	−1.67 (−1.8; −1.45)	−1.61 (−1.80; −1.41)	−0.121 (−0.31; 0.07)	0.21
24-h SBP (mmHg) 39/45	3.13 (−2.14; 8.40)	0.13 (−3.69; 3.95)	5.1 (−0.01; 10.19)	0.05
24-h DBP (mmHg) 39/45	3.92 (0.71; 7.13)	−0.93 (−4.93; 3.06)	3.5 (−0.53; 7.7)	0.09
Daytime SBP (mmHg)	5.02 (0.37; 9.68)	−1.71 (−5.04; 1.62)	4.3 (−1.0; 9.7)	0.11
Daytime DBP (mmHg)	2.77 (−0.15; 5.70)	0.42 (−2.20; 3.04)	1.3 (−2.5; 5.0)	0.5
Nighttime SBP (mmHg)	−0.72 (−9.10; 7.64)	3.87 (−2.92; 10.65)	−0.1 (−9.0; 8.9)	1.0
Nighttime DBP(mmHg)	0.72 (−3.98; 5.42)	3.33(−1.32; 7.98)	−1.5 (−7.1; 4.2)	0.61
Weight (kg) 45/48	−1.32 (−2.10; −0.60)	−0.18 (0.73; 0.36)	−1.1 (−2.0; −0.1)	0.02

* The number of participants analyzed is different between variables due to missing data. ** Intervention delta minus control delta. *** General linear model (GLM) adjusted for respective baseline values. SBP, systolic blood pressure; DBP, diastolic blood pressure.

## Data Availability

The data presented in this study are available on request from the corresponding author.

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
