# Peer review of "Efficacy of an Educational Intervention for Sodium Restriction in Patients with Hypertension: A Randomized Controlled Trial"

_nutrients, 2023, doi:10.3390/nu15092159_

Round 1

Reviewer 1 Report

I read the manuscript with high interest. I have the following minor comments:

Introduction:

Reference 17 is incomplete

Results:

The design of Figure 1 should be modified (the arrows and the text outside the boxes).

The decimal points should be the same throughout the manuscript (not one decimal point as in insignificant p values)

Author Response

REVIEWER 1

Comments and Suggestions for Authors

I read the manuscript with high interest. I have the following minor comments:

Introduction:

Reference 17 is incomplete.

Response – The reference was completed and had been changed to number 18:

“Azjen I. Constructing a TPB Questionnaire: conceptual and methodological.

  1. Accessed May 2013. https://people.umass.edu/aizen/pdf/tpb.measurement.pdf.”

Results:

The design of Figure 1 should be modified (the arrows and the text outside the boxes).

Response - Thank you for the appointment.  Corrected.

The decimal points should be the same throughout the manuscript (not one decimal point as in insignificant p values)

Response - Thank you for the appointment.  Corrected for two decimal points in all the manuscript.

Reviewer 2 Report

Dear authors,

Thanks for the study, the aim was to analyse the efficacy of an educational intervention on sodium consumption and blood pressure is relevant and essential. However, manuscript needs some revisions:

1. Please use the common terminology - either salt or sodium, not both - one term in one sentence, another term in the second.

2. Please use the same unit of measurement for sodium throughout the manuscript, sodium is given in mg in one place, while in another place it is given in g.

3. The introduction is very poor.

4.The introduction of the study lacks a hypothesis.

5. It would be recommended to move Figure 1 and the first paragraph from the results section to the methods section – participants and randomization.

6. It would be recommended to indicate the recommended/preferred reference for all indicators, such as total cholesterol, LDL and HDL cholesterol, Triglycerides, Fasting glucose, Creatinine, Potassium, in Table 1.

7. I would suggest expanding the discussion part.

Author Response

REVIEWER 2 Comments and Suggestions for Authors

Dear authors,

Thanks for the study, the aim was to analyse the efficacy of an educational intervention on sodium consumption and blood pressure is relevant and essential. However, manuscript needs some revisions:

  1. Please use the common terminology - either salt or sodium, not both - one term in one sentence, another term in the second.

Response - the manuscript was revised, and “sodium” was used.

  1. Please use the same unit of measurement for sodium throughout the manuscript, sodium is given in mg in one place, while in another place it is given in g.

Response - Thank you for the appointment. We correct to mg in all text.

  1. The introduction is very poor.

Response - We reviewed and expand the introduction section. Please, see the manuscript.

4.The introduction of the study lacks a hypothesis.

Response - The hypothesis was included: “Our study hypothesized that patients with hypertension who participated in the intervention group would show improved adherence to a low-sodium diet, resulting in decreased blood pressure values assessed by ABPM and urinary sodium excretion values after three and six months compared to the control group. Therefore, the aim of our study was to determine the efficacy of an educational intervention based on DSRQ scores in reducing sodium consumption and lowering BP.”

  1. It would be recommended to move Figure 1 and the first paragraph from the results section to the methods section – participants and randomization.

Response – we moved the figure 1 to the methods section. The initial text in the results was modified: “From 120 participants included, 45 completed follow-up in the intervention group and 50 in the control group. Most were women (67.5%) and Caucasians (68.3%) aged 61.8 ± 10.0 years, and both groups had approximately 18 ± 13 years since the hypertension diagnosis.”

  1. It would be recommended to indicate the recommended/preferred reference for all indicators, such as total cholesterol, LDL and HDL cholesterol, Triglycerides, Fasting glucose, Creatinine, Potassium, in Table 1.

Response – We included the reference values in the footnote of the table 1:

Laboratorial reference values: Total cholesterol <190mg/dL, LDL-C <130mg/dL, HDL-C >40mg/dL, Triglycerides <150mg/dL, Fasting glucose <100mg/dL, Creatinine 0.50 – 1.20 woman and 0.60 – 1.30mg/dL man, Potassium 3.5 – 5.1mEq/L.

  1. I would suggest expanding the discussion part.

Response – We revised the discussion part of the manuscript:

  1. Discussion

The aim of this randomized controlled trial was to compare the effectiveness of an educational intervention that used DSRQ with usual care provided by registered dietitians. After six months, the estimated sodium urinary excretion was similarly reduced in both groups. However, no statistically significant reduction was observed in 24-h ABPM measurements after six months.

The validated version of DSRQ for patients with hypertension from southern Brazil was used for the first time in this randomized clinical trial. At the follow-up, the DSRQ score for the attitude and subjective norm scale showed that participants in the intervention group identified more benefits of following a low-sodium diet compared to the control group. They also reported fewer barriers related to reducing sodium intake, which improved their perceived behavioral control subscale. Additionally, they were motivated to change their behavior and learned skills to prepare low-sodium meals. Participants also reported adapting to the taste of low-sodium foods. However, the dependent behavior subscale showed that eating away from home and choosing a restaurant did not in-fluence participants’ commitment to following the low-sodium diet. Participants re-ported being adapted to the taste of low-sodium food, decreasing sodium amount in meal preparation, and not-adding sodium-based condiments. They preferred to cook and to eat at home; as for food purchased, participants gave preference to not ready-to-eat food: meat, cereals, fruit and vegetables, and no ultra-processed food.

Study on patients with heart failure showed that attitude subscale scores improved after a 6-week intervention, indicating better attitudes toward following a low-sodium diet [26]. Our findings agree with other studies regarding participants perceiving fewer barriers to sodium restriction, improving the ability to read labels, choose foods and eat meals, and reinforcing the perceived ability to deal with possible obstacles [27]. Participants presented positive attitude towards following a low-sodium diet. The knowledge about the low-sodium diet importance to control hypertension may influence dietary intake after explanation.

Our results agree with a study performed on hypertensive adults and the elderly, who had an intake above the recommended, primarily based on sodium added during cooking and at the table [29]. Like our findings, other studies showed that most participants reported receiving advice about sodium restriction, following a prescribed diet, and considering a convenient diary [26, 30, 31]. However, the 24-hour sodium estimated at baseline was higher than the sodium intake goal for patients with hypertension, indicating poor dietary adherence. The same study with heart failure patients (which also applied the DSRQ) [26] showed no difference in sodium intake [] or 24-hour urinary sodium, suggesting that interventions based on the application of DSRQ might not influence sodium excretion beyond that obtained from usual care.

It is worth mentioning that a few participants in our study also participated in a previous study and were seen by different healthcare teams for a considerable period, meaning they were granted access to substantial advice on diet, sodium restriction, and lifestyles. A residual effect of previous experiences might have contributed to both groups' observed sodium intake reduction.

Shamsi et al. reported that a lifestyle intervention in the form of a continuous care model decreased the mean dietary sodium intake and systolic and diastolic BP in hypertensive patients compared with usual care [32]. Unlike our study, their control group received no lifestyle intervention or counseling. In our study, the reduction in sodium intake did not translate to BP reduction. This population frequently uses diuretics which can influence sodium excretion [33]; however, its use was similar between groups.

Our findings corroborate previous interventions based on the Theory of Planned Behavior in Brazilian hypertensive patients to promote low-sodium intake [34]. These interventions resulted in a significant decrease in total sodium addiction to meal preparation; nevertheless, there was no significant reduction in 24-hour sodium urinary excretion. Furthermore, behaviors related to sodium intake involved determinants such as motivational (intention) and hedonic aspects [35]. In other words, low-sodium dietary adherence is complex; it involves aspects such as barriers, skills, knowledge, and motivation to achieve sodium restriction. It is critical to identify dietary adherence factors, including treatment, difficulty in changing eating behaviors, lack of motivation, inadequate knowledge, poor social support, and absence of perceived benefit [19, 36]. Inter-vention strategies to reduce sodium should focus on attitudes and cultural norms to promote dietary change.

The study presented some limitations that deserve consideration. The collection of urine spot can be utilized to estimate sodium intake; however, its accuracy remains a topic of controversy [37- 39]. It may not fully represent all fluctuations in intake over time, and its reliability can be negatively affected using diuretics [37], which are commonly used by this population.

Nevertheless, predictive equations for estimated individual mean 24-hour sodium excretion might be valuable for monitoring sodium intake progress [39–41]. Second, the formula [23] used to estimate sodium intake was developed for the Brazilian population, who present with high sodium intake (4,700 mg of sodium per day) [11]. The formula targeted individuals with renal disease and did not use urinary creatinine and potassium data [25]. Finally, individual loss to follow-up or even those who decided not to remain in the study occurred similarly in both groups. The overall sociodemographic characteristics in the intervention and control groups were similar to the participants who completed the follow-up.

Reviewer 3 Report

The study aims to present the efficacy of an educational intervention based on Dietary Sodium Restriction Questionnaire (DSRQ) scores, throught a randomized parallel controlled trial with a 6-month follow-up. The study can be useful to strenght the knowledge about behavioural interventions to control hypertension.

The paper must be improved in some points before its consideration to publish, specially material and methods section:

ABSTRACT

- The results is not for 120 participants, due to lost participants during the intervention...this should be clarified in the abstract. 

MATERIAL AND METHODS

- include the information about who is responsible for screen the eligible participants;

- include the code about the approval of study by hospital’s institutional review board;

- the description about the intervention applied to "educational intervention group" is very insufficient, it could be more usefull describe better the intervention along the six months. At this moment the description about control group "They received an explanatory leaflet on hypertension with general recommendations. Recommendations included reducing salt intake, the risk of consuming high-sodium foods" and the intervention description is not clear the main differences and to allow its replication.

- In intervention group, the dietary advice includes calories restriction in some cases or not? This should be clearifyed;

- The DSRQ 26-item instrument should be better described in methods, its itens/dimensions that are focused in intervention should be highlithed;

- the formula to convert spot urine in 24h urine excretion should be clearly expressed in methods;

- was used some software to convert dietary recalls into nutrients? this should be clarified;

- on statistical descrotion, explain better what was the methods used to imput missing values and variables used in the intention to treat models.

RESULTS

- "120 were randomized to the intervention groups" should be "120 were randomized to the intervention and control groups",

- "30% were currently on a low-sodium diet" - what are the % distributions between both groups (control and intervention) in relation to "currently on low-sodium diet"? is the difference between groups significant? these data should be added to table 1 because it is very relevant;

- "most participants did not go to restaurants, preferring to cook and eat at home, bought meats, cereals, fruits, and vegetables that were not ready to eat, and did not con-sume ultra-processed foods" - the % of this data and differences betwen both groups should be added to table 1, because the relevance for study.

- Figure 1 - complete the figure - withdrawal of intervention group is incomplete (n=)

- verify all study because it uses sometimes "P" and another times 

"p" to significant differences;

- "Weight reduction was more significant" - what do you mean about more significant (p<0.05)? is higher? this should be clarifyed.

Author Response

REVIEWER 3 Comments and Suggestions for Authors

The study aims to present the efficacy of an educational intervention based on Dietary Sodium Restriction Questionnaire (DSRQ) scores, thought a randomized parallel controlled trial with a 6-month follow-up. The study can be useful to strength the knowledge about behavioral interventions to control hypertension.

 The paper must be improved in some points before its consideration to publish, especially material and methods section:

 ABSTRACT

- The results are not for 120 participants, due to lost participants during the intervention...this should be clarified in the abstract. 

Response – Thank you for the appointment. We included in the abstract “We randomized 120 participants (67.5% women and 68.3% Caucasians), and 25 participants were lost to follow-up.”

We corrected fig 1 and included in the tables the number analyzed per group for each variable, taking into account missing data.

MATERIAL AND METHODS

- include the information about who is responsible for screen the eligible participants.

Response - Included in methods: “The research team comprised trained graduate students from the Nursing and Medicine Schools, as well as two registered dietitians who were postgraduate students. Their responsibilities included recruiting participants, inviting them to participate, administering informed consent, screening for eligibility, and collecting data. Additionally, the registered dietitians applied the DSRQ."

- included the code about the approval of the study by the hospital’s institutional review board;

Response - The information was added to methods: “The hospital’s institutional review board approved the study protocol (protocol number 150496), which was registered at ClinicalTrials.gov (NCT02848690)”

- the description about the intervention applied to "educational intervention group" is very insufficient, it could be more usefull describe better the intervention along the six months. At this moment the description about control group "They received an explanatory leaflet on hypertension with general recommendations. Recommendations included reducing salt intake, the risk of consuming high-sodium foods" and the intervention description is not clear the main differences and to allow its replication.

Response - We detailed the intervention group and control group, below:

Educational intervention group.

                The participants in the educational intervention group were provided with comprehensive guidance and recommendations on how to follow a low-sodium diet. This was carried out by a registered dietitian, who was responsible for overseeing the educational intervention. During their initial consultation, the participants received a dietary plan that emphasized the consumption of fruits, vegetables, low-fat dairy, and low-fat and minimally processed food. Additionally, their daily energy intake was reduced by 500 to 1000 kcal based on their baseline weight. To ensure continued adherence to the low-sodium diet, the participants attended monthly 60-minute face-to-face sessions with the registered dietitians who provided encouragement and motivation.

The DSRQ [22] was applied at the baseline to guide intervention activities and strategies, after three months. and at the end of the follow-up to evaluate participant performance concerning attitudes and barriers associated with sodium restriction. Approaches included promoting the individual’s abilities to achieve goals and developing behavioral changes. They also tracked progress in acquiring skills to overcome barriers and difficulties in adhering to a sodium-restricted diet. Participants developed activities and strategies to increase adherence to the low-sodium diet according to the results of DSRQ subscales. The activities included learning sessions to understand the importance of a low-sodium diet and provide information about food choices such as reading labels, selecting, and cooking with low-sodium items, choosing food in restaurants, and changing food and flavor preferences. Dietary intake was assessed using three-day food records to monitor diet adherence.

The activities were developed according to the instrumental subscales: (i) attitude and subjective norm, participants received an explanation to understand the low-sodium dietary importance to control hypertension and the influence of family and others in choices and food preparation; (ii) perceived behavioral control, they identified barriers to low sodium adherence, such as lack of knowledge, interference with socialization, and lack of food selections; they learned to increase information about food choices, cooking or preparing food without sodium, low-sodium diet shopping, evaluating recipes and making suggestions for changes to low-sodium food, and to read labels; (iii) dependent behavior, they received learning sessions on the amount of sodium in food, including sodium quantity demonstrations, low-sodium food selection, changes in food choices in restaurants, being aware of the need to change both taste and food preferences.

The educational intervention was individualized and based on dietary information provided by the patient. They answered the 24-h dietary recall for monitoring low-sodium diet adherence. The 24-h dietary recall was analyzed by computer software, the DietSys [24], to calculate the sodium content of the patient’s dietary intake. Both groups were followed-up in face-to-face appointments monthly, for six months and requested to not change their physical activity practice.

Control group

Participants allocated to the control group had monthly appointments with registered dietitians, in line with standard care practices. During the first visit, participants received an explanatory leaflet on hypertension and provided general recommendations. These recommendations included reducing sodium intake, avoiding high-sodium foods and alcoholic beverages, and losing weight if a body mass index was greater than 25 kg/m2. The DSRQ [22] was administered at baseline, three months after the randomization, and at the end of the follow-up period. Anthropometric and office blood pressure measurements were performed monthly during 30-minute face-to-face appointments throughout the follow-up period.

Anthropometric and office blood pressure measurements were performed monthly. The 24-h ambulatory blood pressure monitoring was performed at baseline and at the end of the study.

Instruments used for data collection

 The DSRQ is a 26-item instrument validated for patients with hypertension [20]. The first section consists of 11 descriptive, multiple-choice items. The second section includes three subscales using a Likert scale to score each question: i) attitude and subjective norm subscale, ii) perceived behavioral control subscale, and iii) dependent behavior subscale.

The attitude and subjective norm subscale scores range from nine to 40. The highest scores indicate a better attitude toward a low-sodium diet and motivation enhancement following the approval of significant others. Perceived behavioral control subscale scores range from three to 15, with a higher score indicating lower perceived control to follow low-sodium restriction. The dependent behavior subscale scores range from four to 20, with a higher score indicating increased difficulties in adhering to the low-sodium diet.

- In intervention group, the dietary advice includes calories restriction in some cases or not? This should be clarified.

Response - The daily energy intake was reduced by 500 to 1000 kcal according to the participant's weight for intervention, and control group was advised for weight loss if a body mass index > 25 kg/m2. The text was revised: “Additionally, their daily energy intake was reduced by 500 to 1000 kcal based on their baseline weight.”

- The DSRQ 26-item instrument should be better described in methods, its itens/dimensions that are focused in intervention should be highlithed;

Response - We detailed the intervention based on DSRQ:

The activities were developed according to the instrumental subscales: (i) attitude and subjective norm, participants received explanation to understand the low-sodium dietary importance to control hypertension and the influence of family and others in choices and food preparation; (ii) perceived behavioral control, they identified barriers to low sodium adherence, such as lack of knowledge, interference with socialization, and lack of food selections; they learned to increase information about food choices, cooking or preparing food without sodium, low-sodium diet shopping, evaluating recipes and making suggestions for changes to low-sodium food, and to read labels; (iii) dependent behavior, they received learning sessions on the amount of sodium in food, including sodium quantity demonstrations, low-sodium food selection, changes in food choices in restaurants, being aware of the need to change both taste and food preferences.

- the formula to convert spot urine in 24h urine excretion should be clearly expressed in methods;

Response - The information was added to the methods, in the “Outcomes”:

The formula to estimate the 24- hour sodium excretion was developed from a multivariate regression equation coefficient:

 Females: Estimated 24-hour urinary sodium excretion (g/ day) = 0.15 + (weight in kg x 0.03) + (sodium in the urine specimen in g/L x 0.63)

Males: Estimated 24-hour urinary sodium excretion (g/ day) = 0.96 + (weight in kg x 0.03) + (sodium in the urine specimen in g/L x 0.63)”

- was used some software to convert dietary recalls into nutrients? this should be clarified;

Response - The information was included at the end of the intervention description, and we added the respective reference:

The 24-h dietary recall was analyzed by computer software, the DietSys, to calculate the sodium content of the patient’s dietary intake.

22.Rodrigues MP, Khandpur N, Fung TT, Sampson L, Oliveira MRM, Willett WC, Rossato SL. Development of DietSys: A comprehensive food and nutrient database for dietary surveys. J Food Compos Anal. 2021;102(June):104030.

- on statistical description, explain better what was the methods used to imput missing values and variables used in the intention to treat models.

Response - We didn´t impute missing values. We added an explanation in the statistical analysis section:

“The analysis was performed according to the intention-to-treat principle, using PASW Statistics 18® (International Business Machines Corp., New York, USA). Because the number of participants included in each analysis is not equal due to missing data, the n was indicated according.”

RESULTS

- "120 were randomized to the intervention groups" should be "120 were randomized to the intervention and control groups."

Response - Thank you for the appointment. We made the correction in the results:” From 120 participants included, 45 completed the follow-up in the intervention group and 50 in the control group”.

- "30% were currently on a low-sodium diet" - what are the % distributions between both groups (control and intervention) in relation to "currently on low-sodium diet"? is the difference between groups significant? these data should be added to table 1 because it is very relevant;

Response - We added the information to table 1: Following low sodium diet: 19 (31.7%) in the intervention group and 18 (30.0%) in the control group; p=0.84.

- "most participants did not go to restaurants, preferring to cook and eat at home, bought meats, cereals, fruits, and vegetables that were not ready to eat, and did not consume ultra-processed foods" - the % of this data and differences between both groups should be added to table 1, because the relevance for study.

Response – The figures are 34 (56.7%) in the intervention group and 33 (55.0%) in the control group (p=0.85). Added to table 1.

- Figure 1 - complete the figure - withdrawal of intervention group is incomplete (n=)

Response – Thank you for the appointment. The figure 1 was corrected. Please, see the manuscript.

- verify all study because it uses sometimes "P" and another times "p" to significant differences;

Response – Thank you for the appointment. Corrected.

- "Weight reduction was more significant" - what do you mean about more significant (p<0.05)? is higher? this should be clarified.

 Response - We modified the sentence:

“Weight reduction was significant in the intervention group compared with the control group (p = 0.02).”

Reviewer 4 Report

This article shows a well-structured, complete work and offers a pleasant and didactic reading. Congratulations to the authors for their work. I only have a few small observations to make, which I will now go into detail:

- Material and method: the authors say that the design, randomisation and details of the intervention were described in another study and invite the reader to consult that publication. I feel that there should be a description, especially of the study design and perhaps somewhat less detailed description of the interventions.

- Educational intervention group: how was the process of acquiring skills to overcome barriers and difficulties in following the restrictive diet monitored?

- I wonder if any of the groups introduced physical exercise to contribute to the improvement of results or if it was an aspect that was controlled for bias.

Author Response

REVIEWER 4 Comments and Suggestions for Authors

This article shows a well-structured, complete work and offers a pleasant and didactic reading. Congratulations to the authors for their work. I only have a few small observations to make, which I will now go into detail:

 - Material and method: the authors say that the design, randomisation and details of the intervention were described in another study and invite the reader to consult that publication. I feel that there should be a description, especially of the study design, and perhaps a somewhat less detailed description of the interventions.

Response - We included the information in the methods section:

Participants and randomization

Participants were recruited from the outpatient clinic or through media advertisements. Potentially eligible participants were screened, and those who met the criteria were invited to a clinical visit. Eligible participants were men and women aged 40 to 80 years who had a diagnosis of hypertension and were undergoing BP-lowering drug treatment at the clinic and had not been visiting a nutritionist in the previous six months. Participants were invited to participate and, after signing a consent form, were randomly allocated to either the educational intervention or the control group. Exclusion criteria included gastrointestinal tract disease, inflammatory disease, chemotherapy treatment, physician diagnosis of diabetes mellitus, those with cognitive impairment detected during the interview, and those unable to participate without third-party involvement, pregnant or lactating women. The follow-up was carried out for up to six months with monthly monitoring.

Randomization, allocation concealment, and blinding

A randomization list was generated using software (randomization.com) at a 1-to-1 ratio, with participants allocated in blocks of six, and the randomization sequence was created by an independent researcher outside the clinic. The sequence was stored in opaque sealed envelopes which were kept outside the clinical center for blinding purposes.

Participants and the research team were not blinded to the intervention and control groups. However, the assessment of outcomes was conducted by an independent blinded researcher.

Between November 2015 and October 2017, we screened 460 participants. Out of these, 160 did not meet the eligibility criteria, and 180 declined to participate. A total of 120 participants were randomized to the intervention and control groups. By April 2018, 15 and 10 participants were lost to follow-up in the intervention and control groups, respectively. Figure 1 illustrates the screening and enrollment process.

Educational intervention group.

                The participants in the educational intervention group were provided with comprehensive guidance and recommendations on how to follow a low-sodium diet. This was carried out by a registered dietitian, who was responsible for overseeing the educational intervention. During their initial consultation, the participants received a dietary plan that emphasized the consumption of fruits, vegetables, low-fat dairy, and low-fat and minimally processed food. Additionally, their daily energy intake was reduced by 500 to 1000 kcal based on their baseline weight. To ensure continued adherence to the low-sodium diet, the participants attended monthly 60-minute face-to-face sessions with the registered dietitians who provided encouragement and motivation.

The DSRQ [22] was applied at the baseline to guide intervention activities and strategies, after three months and at the end of the follow-up to evaluate participant performance concerning attitudes and barriers associated with sodium restriction. Approaches included promoting the individual’s abilities to achieve goals and developing behavioral changes. They also tracked progress in acquiring skills to overcome barriers and difficulties in adhering to a sodium-restricted diet. Participants developed activities and strategies to increase adherence to the low-sodium diet according to the results of DSRQ subscales. The activities included learning sessions to understand the importance of a low-sodium diet and provide information about food choices such as reading labels, selecting, and cooking with low-sodium items, choosing food in restaurants, and changing food and flavor preferences. Dietary intake was assessed using three-day food records to monitor diet adherence.

The activities were developed according to the instrumental subscales: (i) attitude and subjective norm, participants received an explanation to understand the low-sodium dietary importance to control hypertension and the influence of family and others in choices and food preparation; (ii) perceived behavioral control, they identified barriers to low sodium adherence, such as lack of knowledge, interference with socialization, and lack of food selections; they learned to increase information about food choices, cooking or preparing food without sodium, low-sodium diet shopping, evaluating recipes and making suggestions for changes to low-sodium food, and to read labels; (iii) dependent behavior, they received learning sessions on the amount of sodium in food, including sodium quantity demonstrations, low-sodium food selection, changes in food choices in restaurants, being aware of the need to change both taste and food preferences.

The educational intervention was individualized and based on dietary information provided by the patient. They answered the 24-h dietary recall for monitoring low-sodium diet adherence. The 24-h dietary recall was analyzed by computer software, the DietSys [24], to calculate the sodium content of the patient’s dietary intake. Both groups were followed up in face-to-face appointments monthly, for six months and requested to not change their physical activity practice.

- Educational intervention group: how was the process of acquiring skills to overcome barriers and difficulties in following the restrictive diet monitored?

Response – We added to the educational intervention group description. Please, see above “Educational intervention group.”

- I wonder if any of the groups introduced physical exercise to contribute to the improvement of results or if it was an aspect that was controlled for bias.

Response – We included in the “Trial conduct” section: “Both groups were requested not to change their physical activity practice.”

Round 2

Reviewer 2 Report

Dear authors,

thank you for your revised manuscript. The manuscript is to be published in the journal.

Reviewer 3 Report

accept